# BASELINE-CORRECTED SPACE-BY-TIME NON-NEGATIVE MATRIX FACTORIZATION FOR DECODING SINGLE TRIAL POPULATION SPIKE TRAINS

## ABSTRACT

Activity of populations of sensory neurons carries stimulus information in both the temporal and the spatial dimensions. This poses the question of how to compactly represent all the information that the population codes carry across all these dimensions. Here, we developed an analytical method to factorize a large number of retinal ganglion cells' spike trains into a robust low-dimensional representation that captures efficiently both their spatial and temporal information. In particular, we extended previously used single-trial space-by-time tensor decomposition based on non-negative matrix factorization to efficiently discount pre-stimulus baseline activity. On data recorded from retinal ganglion cells with strong pre-stimulus baseline, we showed that in situations where the stimulus elicits a strong change in firing rate, our extensions yield a boost in stimulus decoding performance. Our results thus suggest that taking into account the baseline can be important for finding a compact information-rich representation of neural activity.

## 1 INTRODUCTION

Populations of neurons encode sensory stimuli across the time dimension (temporal variations), the space dimension (different neuron identities), or along combinations of both dimensions (Buonomano & Maass, 2009; Panzeri et al., 2010; 2015; Harvey et al., 2012; Runyan et al., 2017). Consequently, understanding the neural code requires characterizing the firing patterns along these dimensions and linking them to the stimuli (Abeles & Gerstein, 1988; Haefner et al., 2013; Panzeri et al., 2015; Pouget et al., 2000; Kristan & Shaw, 1997). There are many methods for compactly representing neural activity along their most relevant dimensions. These methods include Principal Component Analysis (PCA), Independent Component Analysis (ICA) and Factor Analysis (FA) (Churchland et al., 2010; Cunningham & Byron, 2014; Laubach et al., 1999; Shen & Meyer, 2008). Recently, a particularly promising tensor decomposition method was introduced that provides a compact representation of single trial neuronal activity into spatial and temporal dimensions and their combination in the given trial (Onken et al., 2016). The method is based on non-negative matrix factorization (NMF) (Lee & Seung, 1999; Devarajan, 2008; Smaragdis et al., 2014) which imposes non-negativity constraints on the extracted components leading to a parts-based, low dimensional, though flexible representation of the data, only assuming non-negativity of the model components. Though space-by-time NMF yielded robust decoding performance with a small number of parameters and good biological interpretability of its basis functions on data recorded from salamander retinal ganglion cells, the method does have a potential shortcoming: it cannot explicitly discount, and is partly confounded by, baseline activity that is not relevant for the neural response to a sensory stimulus. Although these non-negative tensor factorizations performed well on salamander retinal ganglion cells, which have almost non-existent spontaneous activity (Delis et al., 2016), it is not clear how well the method would perform on data with considerable spontaneous activity, which might require to explicitly correct for the pre-stimulus baseline.

One way to reduce the baseline would be to subtract it from the stimulus-elicited response. This, however, would result in negative activities that cannot be modeled using a decomposition with full non-negativity constraints such as space-by-time NMF. In this study, we thus propose a variant of space-by-time NMF that discounts the baseline activity by subtracting the pre-stimulus baseline from each trial and then decomposes the baseline-corrected activity using a tri-factorization that

finds non-negative spatial and temporal modules, and signed activation coefficients. We explored the benefits that this method provides on data recorded from mouse and pig retinal ganglion cells and showed that baseline-corrected space-by-time NMF improves decoding performance on data with non-negligible baselines and stimulus response changes.

## 2 METHODS

We consider data that are composed of trials of spiking activity recorded simultaneously from a population of neurons in response to sensory stimuli. Each trial thus has a temporal component (when is a neuron firing) and a spatial component (which neuron is firing). We aim to find a decomposition into spatial and temporal firing patterns and into coefficients that represent the strength of spatial and temporal firing combinations within a given trial. Before decomposing the data, we discretize neural activity by binning the spike trains into time intervals (chosen to maximize decoding performance, c.f. supplementary Fig. S1) and counting the number of spikes in each bin. We then apply two different tensor decomposition methods that separate the spatial and temporal dimensions. We describe both methods in the following sections.

### 2.1 SPACE-BY-TIME NON-NEGATIVE TENSOR DECOMPOSITION

Following Onken et al. (2016), we decomposed neural activity into spatial and temporal patterns and their activation coefficients. The decomposition of a trial $s$ takes the following form:

$$\mathbf{R}_s = \mathbf{B}_{\text{tem}}\mathbf{H}_s\mathbf{B}_{\text{spa}} + \text{residual} \tag{1}$$

where $\mathbf{R}_s$ denotes the population spike count matrix on trial $s$ across $T$ time bins and $N$ recorded neurons, $\mathbf{B}_{\text{tem}}$ denotes a $(T \times P)$-matrix whose columns are the temporal modules, $\mathbf{B}_{\text{spa}}$ is a $(L \times N)$-matrix whose rows are the spatial modules and $\mathbf{H}_s$ is a $(P \times L)$-coefficient matrix that contains the weights of each combination of spatial and temporal modules. Note that the spatial and temporal modules are trial-independent whereas the activation coefficients $\mathbf{H}_s$ are trial-dependent.

The main goal of the above decomposition is to factorize the input signal into invariant spatial and temporal patterns across trials such that the factorization minimizes the total reconstruction error. Following Onken et al. (2016), we used Space-by-Time Non-negative Matrix Factorization (SbT-NMF, corresponding to the Tucker-2 tensor decomposition with non-negativity constraints) to find the factors of Eq. 1. The algorithm decomposes the input tensor $\mathbf{R}$ into non-negative temporal modules $\mathbf{B}_{\text{tem}}$, non-negative spatial modules $\mathbf{B}_{\text{spa}}$ and non-negative activation coefficient using multiplicative update rule to minimize the Frobenius norm of the difference between the input data and the reconstruction data $\sum_{s=1}^{S} ||\mathbf{R}_s - \mathbf{B}_{\text{tem}}\mathbf{H}_s\mathbf{B}_{\text{spa}}||^2$.

On data recorded from salamander retinal ganglion cells the algorithm was shown to provide low-dimensional data-robust representations of spike trains that capture efficiently both their spatial and temporal information about sensory stimuli (Onken et al., 2016).

### 2.2 BASELINE-CORRECTED SPACE-BY-TIME NON-NEGATIVE TENSOR DECOMPOSITION

To discount baseline activity from neural data, here we propose a novel decomposition algorithm termed Baseline-Corrected Space-by-Time Non-negative Matrix Factorization (BC-SbT-NMF) that first subtracts the baseline from the neural activity and then factorizes the activity into spatial and temporal modules and activation coefficients. Contrary to the original population spike counts, the baseline-corrected data are not necessarily non-negative anymore after baseline subraction. Our decomposition method therefore faces the problem of factorizing signed data. For this purpose, our factorization algorithm decomposes the neural activity into non-negative spatial and temporal modules and signed activation coefficients, corresponding to a Tucker-2 tensor decomposition with non-negativity constrained factor matrices and unconstrained core.

The method is illustrated in Figure 1. Each trial consists of spatio-temporal binned spike counts of a population of neurons. For each neuron, we subtract the pre-stimulus firing rate baseline in the same trial from its activity. We then decompose the baseline-corrected trials into spatio-temporal modules representing common firing patterns and corresponding activation coefficients representing the activity of a particular pattern in a trial. The spatio-temporal patterns in turn are factorized

into spatial modules representing coactive neurons and temporal modules representing temporal population activity patterns.

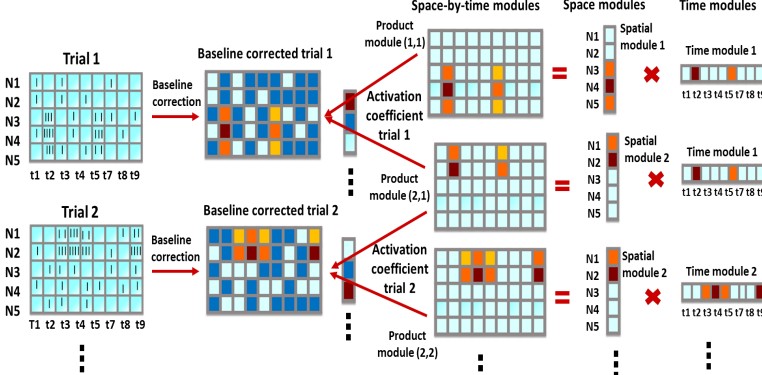

Figure 1: Baseline-corrected space-by-time non-negative matrix factorization. On the left, two trials with spike counts recorded from 5 neurons and 9 time points each are illustrated as a spike count word matrix. The pre-stimulus baseline is subtracted from the spike count word matrix in order to discount the baseline. Then, the signed corrected activity is decomposed into non-negative spatial modules describing coactive neurons, non-negative temporal modules describing the temporal response profile of the population and signed activation coefficient. Each pair of spatial and temporal modules can be combined to form a spatio-temporal module.

The algorithm estimates modules using iterative update rules to minimize the Frobenius norm of the difference between input data and the reconstructed data. Our derivation follows the derivation of semi-NMF presented in  Ding et al. (2010), but extends the derivation to a trial-based tri-factorization where two of the factors are non-negative and trial-independent and one factor is signed and trial-dependent. Relaxation of the constraints for the activation coefficients yields spatial and temporal modules that are less sparse. To counteract this, we also included $L_1$-regularization for the spatial and temporal modules in our derivation following  Hoyer (2004).

Our objective function takes the form:

$$
\begin{aligned}
E^2 &= \sum_{s=1}^{S} ||\mathbf{R}_s - \mathbf{B}_{\text{tem}}\mathbf{H}_s\mathbf{B}_{\text{spa}}||^2 + \lambda \sum_{i,k}[\mathbf{B}_{\text{tem}}]_{ik} + \lambda \sum_{i,k}[\mathbf{B}_{\text{spa}}]_{ik} \quad (2) \\
&= \sum_{s=1}^{S} \text{Tr}\left(\mathbf{R}_s^{\mathsf{T}}\mathbf{R}_s - 2\mathbf{R}_s^{\mathsf{T}}\mathbf{B}_{\text{tem}}\mathbf{H}_s\mathbf{B}_{\text{spa}} + \mathbf{B}_{\text{spa}}^{\mathsf{T}}\mathbf{H}_s^{\mathsf{T}}\mathbf{B}_{\text{tem}}^{\mathsf{T}}\mathbf{B}_{\text{tem}}\mathbf{H}_s\mathbf{B}_{\text{spa}}\right) \quad (3) \\
&\quad + \lambda \sum_{i,k}[\mathbf{B}_{\text{tem}}]_{ik} + \lambda \sum_{i,k}[\mathbf{B}_{\text{spa}}]_{ik},
\end{aligned}
$$

where $\lambda \geq 0$ denotes the regularization parameter. For simplicity, we used the same parameter for the spatial and the temporal modules, but one could also have separate parameters.

We derived iterative update steps for $\mathbf{B}_{\text{spa}}$, $\mathbf{B}_{\text{tem}}$ and $\mathbf{H}$ to minimize our objective function.

We started with the derivation of an iterative update step for $\mathbf{B}_{\text{spa}}$. To cast the tensor factorization of the $(T \times N \times S)$ tensor $\mathbf{R}$ as a matrix tri-factorization, we first reshaped $\mathbf{R}$ by concatenating all slices $\mathbf{R}_s$ along the first tensor dimension. This yields a data matrix $\mathbf{R}_{\text{spa}}$ of size $(ST \times N)$. Analogously, we represented the $(P \times L \times S)$ tensor $\mathbf{H}$ of activation coefficients as a $(SP \times L)$ matrix $\mathbf{H}_{\text{spa}}$.

This procedure allows us to formulate a single Lagrangian with regard to $\mathbf{B}_{\text{spa}}$:

$$
\mathcal{L}(\mathbf{B}_{\text{spa}}) = \text{Tr}\left(-2\mathbf{R}_{\text{spa}}^{\mathsf{T}}\mathbf{B}_{\text{tem}}\mathbf{H}_{\text{spa}}\mathbf{B}_{\text{spa}} + \mathbf{B}_{\text{spa}}^{\mathsf{T}}\mathbf{H}_{\text{spa}}^{\mathsf{T}}\mathbf{B}_{\text{tem}}^{\mathsf{T}}\mathbf{B}_{\text{tem}}\mathbf{H}_{\text{spa}}\mathbf{B}_{\text{spa}} - \beta\mathbf{B}_{\text{spa}}\right) + \lambda \sum_{i,k}[\mathbf{B}_{\text{spa}}]_{ik} \quad (4)
$$

where the Lagrangian multipliers $\beta_{ij}$ enforce non-negativity constraints $[\mathbf{B}_{\mathrm{spa}}]_{ij} \geq 0$. With the zero-gradient condition, we get:

$$\frac{\partial \mathcal{L}}{\partial \mathbf{B}_{\mathrm{spa}}} = -2\mathbf{R}_s^{\mathsf{T}}\mathbf{B}_{\mathrm{tem}}\mathbf{H}_{\mathrm{spa}} + 2\mathbf{B}_{\mathrm{spa}}^{\mathsf{T}}\mathbf{H}_s^{\mathsf{T}}\mathbf{B}_{\mathrm{tem}}^{\mathsf{T}}\mathbf{B}_{\mathrm{tem}}\mathbf{H}_{\mathrm{spa}} - \beta + \lambda = 0 \tag{5}$$

The complementary slackness condition then yields:

$$\left(-2\mathbf{R}_{\mathrm{spa}}^{\mathsf{T}}\mathbf{B}_{\mathrm{tem}}\mathbf{H}_{\mathrm{spa}} + 2\mathbf{B}_{\mathrm{spa}}^{\mathsf{T}}\mathbf{H}_{\mathrm{spa}}^{\mathsf{T}}\mathbf{B}_{\mathrm{tem}}^{\mathsf{T}}\mathbf{B}_{\mathrm{tem}}\mathbf{H}_{\mathrm{spa}} + \lambda\right)_{ik}[\mathbf{B}_{\mathrm{spa}}]_{ik} = \beta_{ik}[\mathbf{B}_{\mathrm{spa}}]_{ik} = 0 \tag{6}$$

In this equation, the parts $\mathbf{R}_{\mathrm{spa}}^{\mathsf{T}}\mathbf{B}_{\mathrm{tem}}\mathbf{H}_{\mathrm{spa}}$ and $\mathbf{H}_{\mathrm{spa}}^{\mathsf{T}}\mathbf{B}_{\mathrm{tem}}^{\mathsf{T}}\mathbf{B}_{\mathrm{tem}}\mathbf{H}_{\mathrm{spa}}$ are signed. To derive update rules for the non-negative spatial modules, we therefore need to separate positive and negative parts of the equation. For a matrix $\mathbf{A}$, we denoted its positive and negative parts as $A_{ik}^+ = (|A_{ik}| + A_{ik})/2$ and $A_{ik}^- = (|A_{ik}| - A_{ik})/2$. With this definition, the identity $\mathbf{A} = \mathbf{A}^+ - \mathbf{A}^-$ holds. Separating the positive and negative parts of Eq. 6 by means of this identity, we obtained:

$$[-2((\mathbf{R}_{\mathrm{spa}}^{\mathsf{T}}\mathbf{B}_{\mathrm{tem}}\mathbf{H}_{\mathrm{spa}})^+ - (\mathbf{R}_{\mathrm{spa}}^{\mathsf{T}}\mathbf{B}_{\mathrm{tem}}\mathbf{H}_{\mathrm{spa}})^-) + \lambda$$
$$+ 2((\mathbf{B}_{\mathrm{spa}}^{\mathsf{T}}\mathbf{H}_{\mathrm{spa}}^{\mathsf{T}}\mathbf{B}_{\mathrm{tem}}^{\mathsf{T}}\mathbf{B}_{\mathrm{tem}}\mathbf{H}_{\mathrm{spa}})^+ - (\mathbf{B}_{\mathrm{spa}}^{\mathsf{T}}\mathbf{H}_{\mathrm{spa}}^{\mathsf{T}}\mathbf{B}_{\mathrm{tem}}^{\mathsf{T}}\mathbf{B}_{\mathrm{tem}}\mathbf{H}_{\mathrm{spa}})^-)]_{ik}[\mathbf{B}_{\mathrm{spa}}]_{ik} = \beta_{ik}[\mathbf{B}_{\mathrm{spa}}]_{ik} = 0 \tag{7}$$

At convergence, we have $\mathbf{B}_{\mathrm{spa}}^{(\infty)} = \mathbf{B}_{\mathrm{spa}}^{(t+1)} = \mathbf{B}_{\mathrm{spa}}^{(t)}$. Hence, we obtain the following update step for $\mathbf{B}_{\mathrm{spa}}$:

$$[\mathbf{B}_{\mathrm{spa}}]_{ij} \leftarrow [\mathbf{B}_{\mathrm{spa}}]_{ij}\sqrt{\frac{[(\mathbf{R}_{\mathrm{spa}}^{\mathsf{T}}\mathbf{B}_{\mathrm{tem}}\mathbf{H}_{\mathrm{spa}})^+]_{ij} + [\mathbf{B}_{\mathrm{spa}}^{\mathsf{T}}(\mathbf{H}_{\mathrm{spa}}^{\mathsf{T}}\mathbf{B}_{\mathrm{tem}}^{\mathsf{T}}\mathbf{B}_{\mathrm{tem}}\mathbf{H}_{\mathrm{spa}})^-]_{ij}}{[(\mathbf{R}_{\mathrm{spa}}^{\mathsf{T}}\mathbf{B}_{\mathrm{tem}}\mathbf{H}_{\mathrm{spa}})^-]_{ij} + [\mathbf{B}_{\mathrm{spa}}^{\mathsf{T}}(\mathbf{H}_{\mathrm{spa}}^{\mathsf{T}}\mathbf{B}_{\mathrm{tem}}^{\mathsf{T}}\mathbf{B}_{\mathrm{tem}}\mathbf{H}_{\mathrm{spa}})^+]_{ij} + \lambda}} \tag{8}$$

To derive the update step for $\mathbf{B}_{\mathrm{tem}}$, we analogously reshaped $\mathbf{R}$ by concatenating all slices $\mathbf{R}_s$ along the second tensor dimension to get a data matrix $\mathbf{R}_{\mathrm{tem}}$ of size $(T \times SN)$ and we represented the $(P \times L \times S)$ tensor $\mathbf{H}$ of activation coefficients as a $(P \times SL)$ matrix $\mathbf{H}_{\mathrm{tem}}$. The analogous derivation steps as for the spatial modules then yield the update step for $\mathbf{B}_{\mathrm{tem}}$:

$$[\mathbf{B}_{\mathrm{tem}}]_{ij} \leftarrow [\mathbf{B}_{\mathrm{tem}}]_{ij}\sqrt{\frac{[(\mathbf{H}_{\mathrm{tem}}\mathbf{B}_{\mathrm{spa}}\mathbf{R}_{\mathrm{tem}}^{\mathsf{T}})^+]_{ij} + [\mathbf{B}_{\mathrm{tem}}(\mathbf{H}_{\mathrm{tem}}\mathbf{B}_{\mathrm{spa}}\mathbf{B}_{\mathrm{spa}}^{\mathsf{T}}\mathbf{H}_{\mathrm{tem}}^{\mathsf{T}})^-]_{ij}}{[(\mathbf{H}_{\mathrm{tem}}\mathbf{B}_{\mathrm{spa}}\mathbf{R}_{\mathrm{tem}}^{\mathsf{T}})^-]_{ij} + [\mathbf{B}_{\mathrm{tem}}(\mathbf{H}_{\mathrm{tem}}\mathbf{B}_{\mathrm{spa}}\mathbf{B}_{\mathrm{spa}}^{\mathsf{T}}\mathbf{H}_{\mathrm{tem}}^{\mathsf{T}})^+]_{ij} + \lambda}} \tag{9}$$

Finally, we updated the activation coefficients $\mathbf{H}$ on a trial-by-trial basis. The activation coefficients are signed. For this reason, we can easily obtain the optimal activation coefficient matrices by inverting the module matrices. For each $s \in \{1, \ldots, S\}$, we let

$$\mathbf{H}_s \leftarrow \mathbf{B}_{\mathrm{tem}}^{-1}\mathbf{R}_s\mathbf{B}_{\mathrm{spa}}^{-1} \tag{10}$$

where $(\cdot)^{-1}$ denotes the Moore-Penrose pseudo-inverse.

The complete baseline-corrected space-by-time NMF algorithm takes the following form:

1. For each neuron $i$, calculate the pre-stimulus firing rate $b_i$ and subtract it from the corresponding elements of the data tensor: $\forall i \in \{1, \ldots, N\} : \mathbf{R}_{:,i,:} \rightarrow \mathbf{R}_{:,i,:} - b_i$.

2. Initialize $\mathbf{B}_{\mathrm{tem}}$ $(T \times P)$, $\mathbf{B}_{\mathrm{spa}}$ $(L \times N)$ with non-negative random numbers uniformly distributed between 0 and 1, and $\mathbf{H}$ $(P \times L \times S)$ with random numbers uniformly distributed between -1 and 1.

3. Given $\mathbf{H}, \mathbf{B}_{\mathrm{tem}}$ and the data matrix $\mathbf{R}$ $(T \times N \times S)$, update $\mathbf{B}_{\mathrm{spa}}$:
   (a) Reshape $\mathbf{R} \rightarrow \mathbf{R}_{\mathrm{spa}}$ $(ST \times N)$ and $\mathbf{H} \rightarrow \mathbf{H}_{\mathrm{spa}}$ $(SP \times L)$.
   (b) Update $\mathbf{B}_{\mathrm{spa}}$ using Eq.8

4. Given $\mathbf{H}, \mathbf{B}_{\mathrm{spa}}$ and the data matrix $\mathbf{R}$ $(T \times N \times S)$, update $\mathbf{B}_{\mathrm{tem}}$:
   (a) Reshape $\mathbf{R} \rightarrow \mathbf{R}_{\mathrm{tem}}$ $(T \times SN)$ and $\mathbf{H} \rightarrow \mathbf{H}_{\mathrm{tem}}$ $(P \times LN)$.

(b) Update $\mathbf{B}_{\text{tem}}$ using Eq.9.

5. Given $\mathbf{B}_{\text{tem}}$ and $\mathbf{B}_{\text{spa}}$:

   (a) For all $s \in 1, ..., S$, update $\mathbf{H}_s$ using Eq.10.

6. If decrease in approximation error $\sum_{s=1}^{S} ||\mathbf{R}_s - \mathbf{B}_{\text{tem}}\mathbf{H}_s\mathbf{B}_{\text{spa}}||^2$ is below a given tolerance, stop. Otherwise, go to step 3.

We provide MATLAB code for this algorithm at the following GitHub repository (TO BE ADDED TO THE FINAL VERSION UPON ACCEPTANCE).

## 2.3 DECODING ANALYSIS

To evaluate the performance of the factorization algorithms to extract an informative low-dimensional representation of the neural data, we used multi class linear discriminant analysis (LDA) applied to the activation coefficients $\mathbf{H}_s$ (non-negative in the case of space-by-time NMF and signed in the case of baseline-corrected space-by-time NMF) as predictors.

For each experiment, we randomly separated trials into two sets of equal size: training set and test set and then applied the decomposition methods to the training set to obtain non-negative spatial and temporal modules and related activation coefficient. For fixed spatial and temporal modules, we then also applied the decomposition methods to the corresponding test set trials to compute activation coefficient given the training set modules. Finally, for each experiment and each decomposition method, we trained LDA classifiers on the training set activation coefficients and evaluated the test set decoding performance on the test set activation coefficients.

The decomposition methods have three free parameters: the number of spatial modules, the number of temporal modules and the $L_1$-regularization parameter $\lambda$. We decomposed the data and evaluated decoding performance for each possible combination of the module number parameters, where we used as the maximum number of spatial module the number of recorded neurons and we used as the maximum number of temporal module the number of time bins per trial, and at first setting the regularization parameter to zero. For each experiment and each decomposition method, we then selected the pair of spatial and temporal module numbers that maximized decoding performance. Whenever we found more than one pair that reached optimal decoding performance, we selected the pair with the minimum sum of module numbers. For these optimal module numbers, we then explored the effect of $\lambda$ and did not find an increase in decoding performance for any non-zero $\lambda$ Fig. S3). For this reason, we report all results for $\lambda = 0$.

## 2.4 EXPERIMENTAL SETUP AND DATASETS

We used multi-electrode arrays to record activity from populations of retinal ganglion cells of two mouse retinas (retinas 1 and 2) and one pig retina (retina 3) in scotopic light-level conditions. The experimental procedures were in accordance with international standards for animal research and were approved by the local ethical commitee and the local authorities, with experimental procedures following what detailed in a previous publication (DETAILS OF ETHICAL APPROVAL AND CITATION WILL BE PROVIDED UPON ACCEPTANCE). We recorded simultaneously from 30, 43 and 56 neurons, respectively, using the following five stimulation protocols:

*Natural movies, dance and mouse (NMD, NMM)*: Two kinds of black and white movies were used for visual stimulation. The first one was a natural movie from the viewpoint of the mouse and the second movie was a clip showing a dancing couple. The dance movie was projected 44 times for one retina while the movie from the viewpoint of mouse was projected 44 times for the two other retinas.

*Full-field high and low contrast flicker (HC, LC)* : Screen-brightness values were picked from a Gaussian distribution with mean 30 and standard deviation of 9 for high contrast and 3.5 for low contrast flicker with a range of values between 0 and 60 for hight contrast and a range of values between 20 and 40 for low contrast flicker. Both stimuli were projected 10 times for three retina.

*Full-field steps of light (SoL)* : Homogeneous illuminated screen with the following brightness sequence was presented: gray (pixel value 30) – white (pixel value 50) – gray (pixel value 30) – black (pixel value 10) – gray (pixel value 30). This stimulus was projected 10 times for three retinas.

To define stimulus classes, we cut each continuous stimulus into 1 second intervals and associated a running number with each interval. We then randomly divided each dataset by putting half of the trials of each class into a training trials to compute the decompositions modules and to train the decoder, and the other half of the trials into a test set to evaluate the decoding performance.

# 3 RESULTS

To evaluate the benefits of taking the pre-stimulus baseline firing rate into account when looking for trial-invariant spatial and temporal firing patterns, we first discretized spike train with the time interval that gives highest decoding performance and then decoded retinal ganglion cell responses with and without explicitly taking the baseline into account. More precisely, we obtained low-dimensional representations of mammalian ganglion cell activity using space-by-time NMF and baseline-corrected space-by-time NMF and then used an LDA-classifier to decode the visual stimuli from these representations (c.f. Section Methods).

## 3.1 SPACE-BY-TIME DECOMPOSITION OF DATA RECORDED FROM MAMMALIAN RETINAL GANGLION CELLS

We first explored the low-dimensional representations of the neural populations responses that SbT-NMF and BC-SbT-NMF found. Both methods identified non-negative spatial and temporal modules. The activation coefficients, however, were non-negative only in the case of SbT-NMF, and signed in the case of BC-SbT-NMF.

Figure 2 shows example modules obtained from both decomposition methods. We scaled the vector length of each module to unit length to facilitate visual comparison of the modules. The spatial modules, describing simultaneous firing of groups of neurons, show ensembles of neurons that were coactive during flicker stimulation (Figure 2a, d). The temporal modules, describing temporal population activity profiles, show elongated temporal periods of population activity over the 1 second intervals relative to the onset time of the stimulus class (Figure 2b, e). Notably, the SbT-NMF modules are considerably sparser than the BC-SbT-NMF modules, but otherwise have somewhat similar shapes.

Figure 2 panels c and f show examples of activation coefficients of 4 classes and 2 trials each. These matrices form compact dimensionality-reduced representations of the trials: the number of coefficients per trials ($10 \times 10$) is much smaller than the original number of population spike counts per trial ($43 \times 100$). Visually, one can appreciate that the two trials of a class are more similar (vertical similarity) than trials between classes (horizontal similarity). In the next section, we will make this statement more formal by training a classifier on the activation coefficients and evaluating decoding performance on a separate test set.

## 3.2 DECODING VISUAL INFORMATION FROM RETINAL GANGLION CELLS

We applied an LDA decoder on single trial activation coefficients to evaluate how much information the coefficients carried about the visual stimuli. This allowed us to understand how well the decomposition methods could identify low-dimensional representations of the retinal ganglion cell activity that preserved the relevant features. To avoid overfitting, we evaluated decoding performance on a separate test set.

In addition, we also used spatiotemporal PCA, ICA and orthogonal Tucker-2 [12] to obtain other low-dimensional representations of single trial activity subject to different constraints. However, we found that average test set decoding performances of each of these methods were below that of BC SbT-NMF and SbT-NMF (see supplementary Fig. S2).

Figure 3a and b show the test set classification performance that we obtained for SbT-NMF and BC-SbT-NMF evaluated in five stimulus conditions (NND/NMM, HC, SoL and LC, see Section Methods) on a total of three retinas. Both algorithms captured more information for the movie and high contrast stimuli (NND/NNM, HC; shown in Figure 3 panel a, decoding performance greater than 50% on all datasets) than for the step of light and low contrast (SoL, LC; shown in Figure 3 panel b, decoding performance below 30% on all datasets) stimuli. Within these categories, sometimes SbT-NMF achieved higher decoding performance and other times, BC-SbT-NMF performed better.

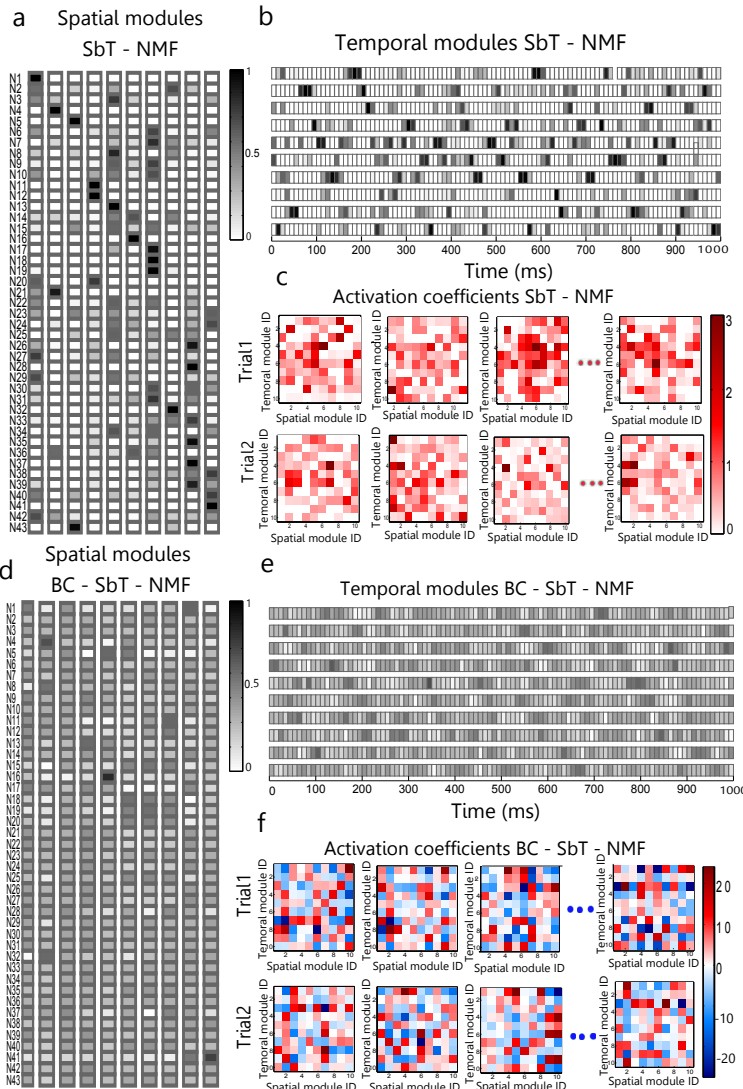

Figure 2: Examples of spatial and temporal modules and activation coefficients identified by SbT-NMF and BC-SbT-NMF on data recorded simultaneously from 43 retinal ganglion cells during low contrast flicker stimulation. (a) Ten non-negative spatial modules (one module per column). (b) Ten non-negative temporal modules (one module per row). (c) Two trials of four classes of activation coefficient matrices containing coefficients for all pairs of spatial and temporal modules. (d)-(f) Like a-c, but for BC-SbT-NMF.

To understand the conditions under which one or the other decomposition method performs better, we investigated the stimulus-elicited change in firing rate from the pre-stimulus baseline.

When evaluating the decoding performance as a function of this change in firing rate, we found that BC-SbT-NMF tended to perform better for high rate changes (Figure 3c). We quantified this (Figure 3d) by calculating the difference in decoding performance between BC-SbT-NMF and SbT-NMF for low rate changes (change $< 0.7$ Hz) and for high rate changes (change $\geq 0.7$ Hz). This split separated the rate changes into two clusters (see Figure 3e).

For low rate changes, we found that SbT-NMF leaned towards performance than BC-SbT-NMF. The difference in decoding performance, however, was not significant (one-tailed $t$-test, $p = 0.0775$). For high rate changes, on the other hand, we found a significant performance increase for BC-SbT-NMF compared to SbT-NMF (one-tailed $t$-test, $p = 0.0438$).

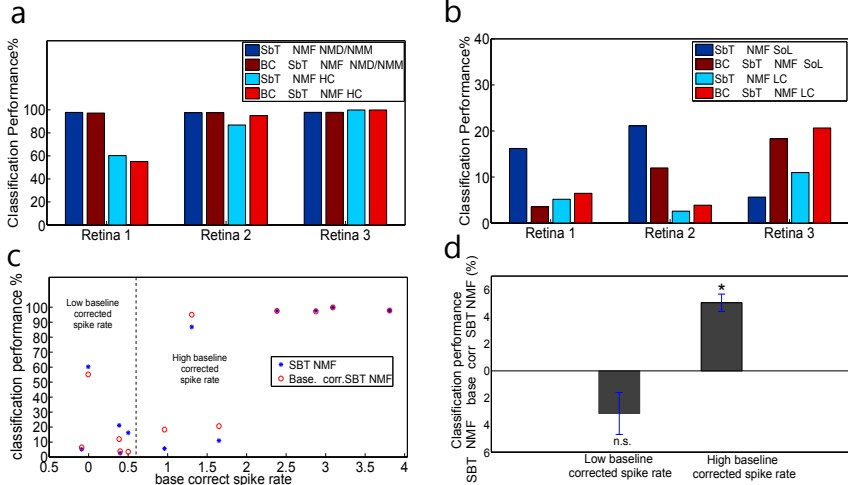

Figure 3: (a) Decoding performance of SbT-NMF (dark and light blue) and BC-SbT-NMF (dark and light red) decompositions of spiking activity recorded from three retinas during natural movie (dark blue and dark red, NMD/NMM) and high contrast flicker (light blue and light red, HC) stimulation. (b) Same as a, but for step of light (SoL) and low contrast flicker (LC) stimuli. (c) Baseline-corrected firing rate vs. decoding performance for SbT-NMF (blue) and BC-SbT-NMF (red) across all visual stimuli. (d) Classification performance difference between BC-SbT-NMF and SbT-NMF for high baseline-corrected firing rates and for low baseline-corrected firing rates. * $p < 0.05$; one-tailed $t$-test. Error bars indicate s.e.m.

In summary, these results show that the BC-SbT-NMF decomposition method can achieve a significant improvement in decoding performance when there is a substantial change from pre-stimulus baseline rate to stimulus-elicited firing rate.

## 3.3 DISCUSSION

Here we introduced a novel computational approach to decompose single trial neural population spike trains into a small set of trial-invariant spatial and temporal firing patterns and into a set of activation coefficients that characterize single trials in terms of the identified patterns. To this end, we extended space-by-time non-negative matrix factorization to discount the neuronal pre-stimulus baseline activity. Subtraction of the baseline required the introduction of signed activation coefficients into the decomposition algorithm. This extension considerable widens the scope of applicability of the algorithm as it opens the possibility to decompose data that are inherently signed.

Our method inherits many the advantages of the original space-by-time NMF decomposition such as yielding low-dimensional representations of neural activity that compactly carry stimulus information from both the spatial and temporal dimension. Using non-negativity constraints for the spatial and temporal modules, we could also retain the ability of space-by-time NMF to identify a parts-based representation of the concurrent spatial and temporal firing activity of the population. The factorization into space and time further still allows the quantification of the relative importance of these different dimensions on a trial-by-trial basis.

Recently, Delis et al. (2016) introduced another tensor decomposition algorithm with the capacity to factorize signed data. Their algorithm differs from ours in that it introduces additional constraints for the spatial and temporal modules (cluster-NMF). Our algorithm, on the other hand, introduces no additional constraints, thereby facilitating the comparison with the original space-by-time NMF algorithm. In fact, our extension actually relaxes the non-negativity constraint for the activation coefficients without giving up the parts-based representation of the spatial and temporal modules. This made it possible to pinpoint the reason for the increase in performance as the introduction of the baseline-correction.

While BC-SbT-NMF outperformed SbT-NMF overall on tasks with strong baseline activity, we also found that in a few cases, SbT-NMF performed better than BC-SbT-NMF. Previous studies showed that there is an effect of the baseline firing rate on the response (Destexhe et al., 2003; Gutnisky et al., 2017). In these situations, the baseline might have an advantageous effect on the representation of neural responses and could lead to better decoding performance of SbT-NMF that we observed in some cases. One possibility to take this effect into account would be to devise a joint factorization-decoding framework that explicitly introduces the baseline into the optimization framework. While this is beyond the scope of the current work, we believe that development of such a framework is a promising direction for future research.

In order to evaluate decoding performance, we applied LDA classification to the single trial activation coefficients to predict the stimulus identity and also to compare decoding performance of our baseline correction extension with the original space-by-time NMF decomposition. Specifically, we could show that our baseline-corrected version of space-by-time NMF increases decoding performance significantly when the difference between pre-stimulus baseline activity and stimulus-elicited rate was moderate to high. Importantly, this rate-change criterion makes it possible to select the best decomposition method (SbT-NMF vs. BC-SbT-NMF) following a simple data screening by means of the rate change. On our data, we obtained a relative difference in decoding performance on the order of 19.18% when picking the right method in this way and comparing to the inferior method.

The requirement for such a rate change to perform well can be understood when considering the baseline-corrected activity. Without a substantial change from pre-stimulus to stimulus-elicited rate, most of the baseline-corrected activity will be close to zero. The Frobenius norm that is at the core of our objective function puts emphasis on high values and will be sensitive to outliers whenever most of the activity is close to zero. In this situation, our update rules are strongly affected by noise, thereby decreasing cross-validated decoding performance. In practical terms, this new method is expected to improve decoding performance when there is a large sensory-evoked response but the differences in responses across different sensory stimuli is of the order of spontaneous activity. In that case, the discounting of the spontaneous levels of firing would help to better discriminate among different stimuli based on neural responses. While the original space-by-time NMF algorithm could in principle identify spatial and temporal modules that fully account for the implicit baseline, the performance gain of our extension suggests that in practice, the original method cannot completely do so. Additional modules increases the model complexity and the number of parameters the method needs to fit which lowers decoding performance. The discount of the baseline provides an elegant way to avoid this unnecessary complication.

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

## SUPPORTING INFORMATION

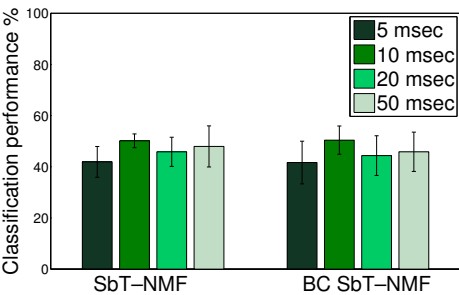

Figure S1: Decoding performance of SbT–NMF and BC–SbT–NMF on discretized spike trains with four different time resolutions (5 ms, 10 ms, 20 ms and 50 ms) recorded from three retinas during presentation of natural movie, high contrast flicker, step of light and low contrast flicker stimuli.

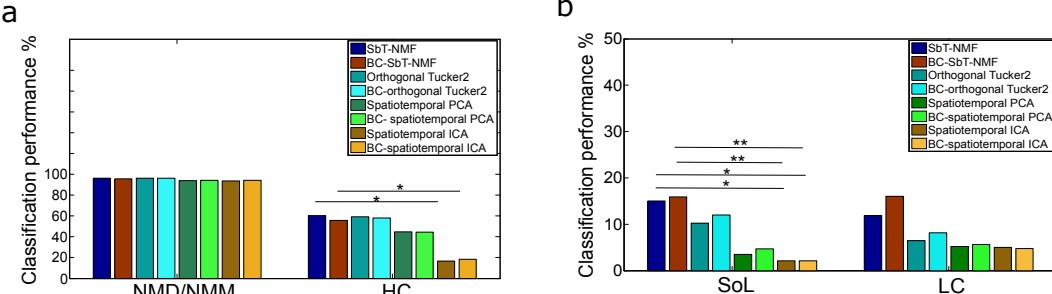

Figure S2: Comparison of decoding performance of various methods. (a) Decoding performance of SbT-NMF, BC-SbT-NMF, orthogonal Tucker2, BC-orthogonal Tucker 2, spatiotemporal PCA, BC-spatiotemporal PCA, spatiotemporal ICA and BC-spatiotemporal ICA on spiking activities recorded from three retinas during presentation of natural movie (NMD/NMM) and high contrast flicker (HC) stimuli. (b) Same as a, but for step of light (SoL) and low contrast flicker (LC) stimuli.

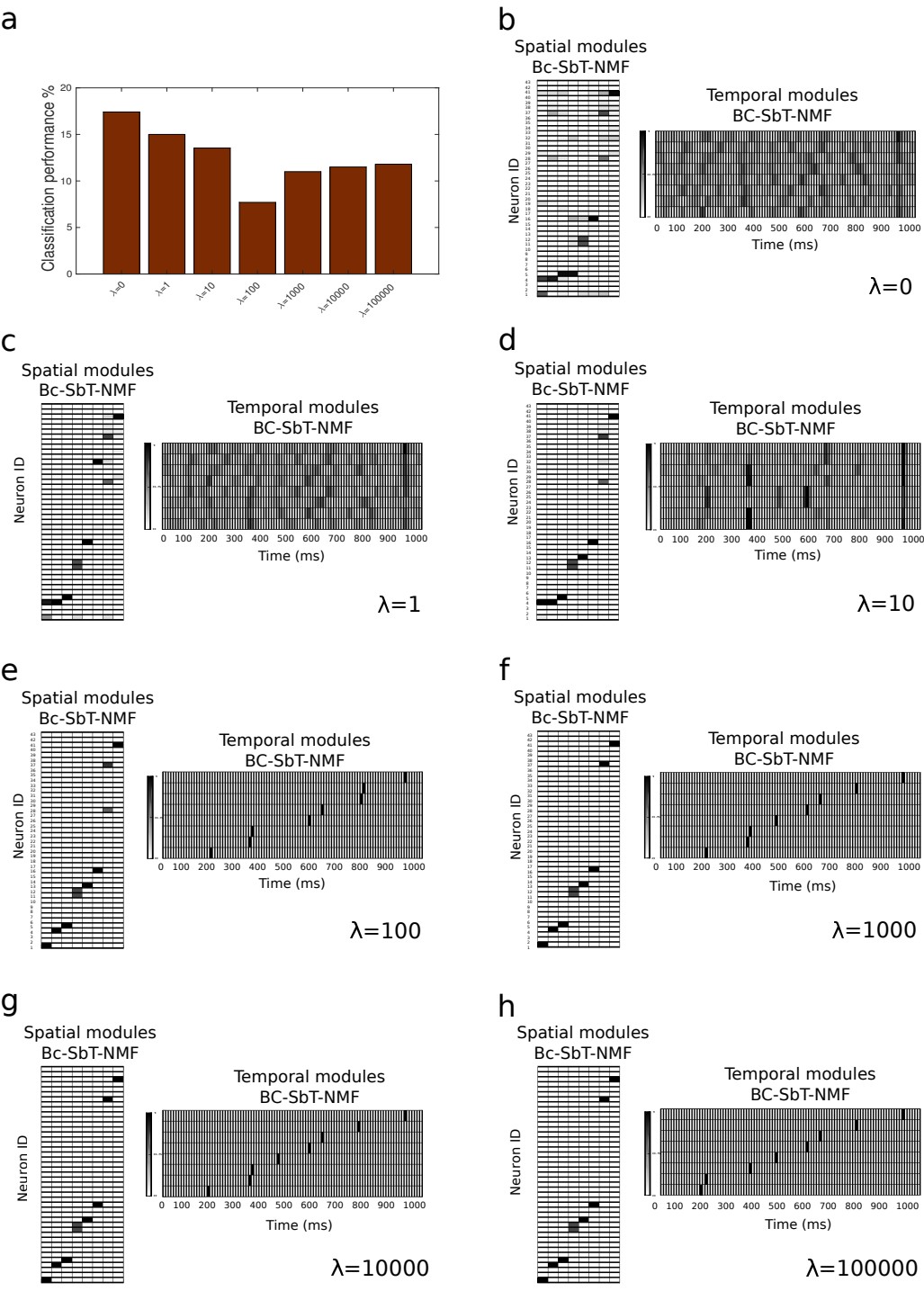

Figure S3: $L_1$-regularization on BC-SbT-NMF spatial and temporal modules obtained from step of light stimuli (SoL). (a) Decoding performance of BC-SbT–NMF regularized with 0, 1, 10, 100, 1000, 10000 and 100000 sparsity constraints. (b-h) Examples of spatial and temporal modules identified by BC-SbT-NMF regularized with $\lambda = 0$ (b), $\lambda = 1$ (c), $\lambda = 10$ (d), $\lambda = 100$ (e), $\lambda = 1000$ (f), $\lambda = 10000$ (g) and $\lambda = 100000$ (h).

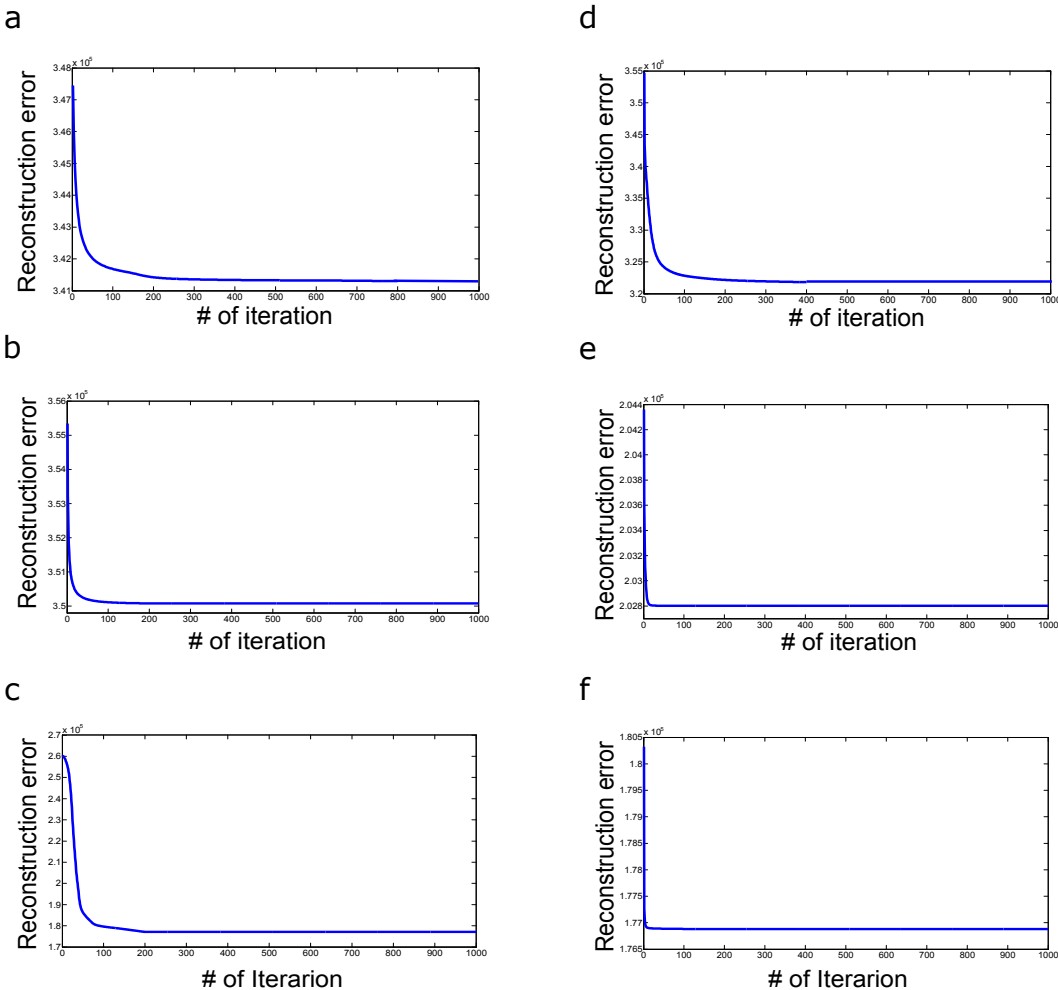

Figure S4: Convergence of multiplicative update rules. (a-c) Reconstruction error of SbT-NMF decomposition as a function of update rule iteration during (a) Natural Movie (NMM), (b) High Contrast flicker (HC) and (c) Low Contrast flicker (LC) stimuli. (d-f) Same as (a-c) but for BC-SbT-NMF decomposition.

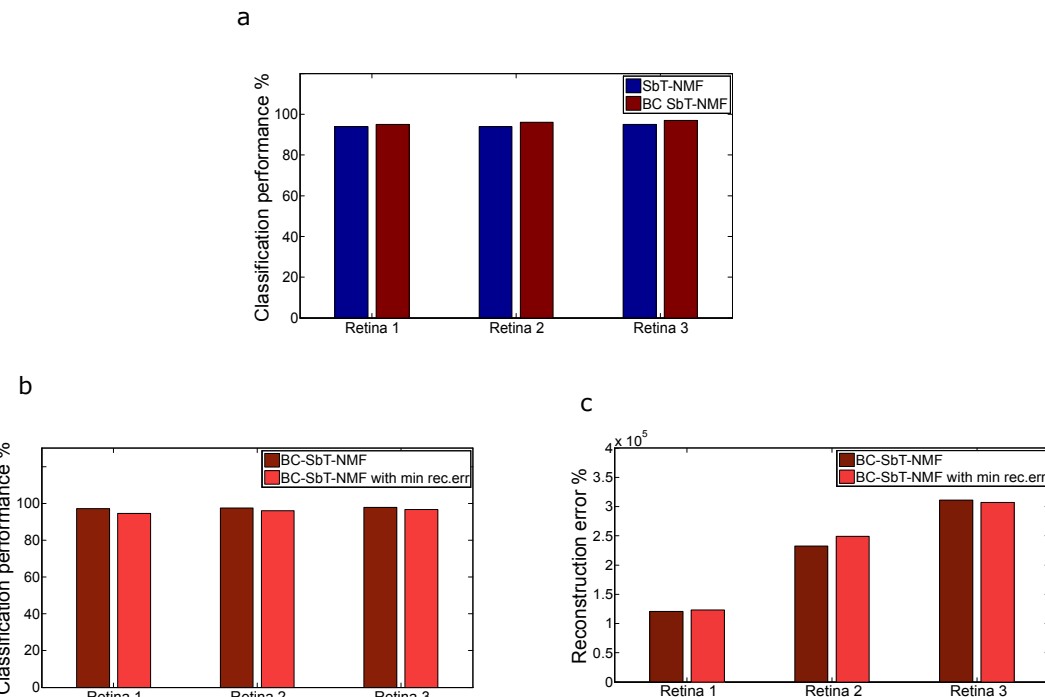

Figure S5: Evaluation of multiple decompositions with random initializations to avoid local minima. (a) Decoding performance of SbT-NMF and BC-SbT-NMF when running 50 decompositions with different random initializations and selecting the decomposition with the smallest reconstruction error for three retinas and presentation of natural movie (NMD/NMM). (b) Change in decoding performance of BC-SbT-NMF with single decomposition and with smallest reconstruction error. (c) Change in reconstruction error of BC-SbT-NMF with single decomposition and with smallest reconstruction error.

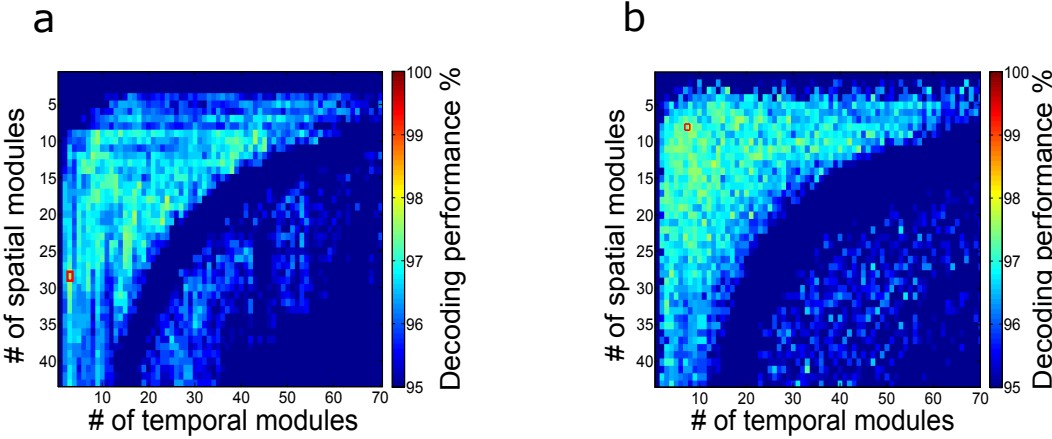

Figure S6: Effect of number of modules on decoding performance. Training set decoding performance of (a) SbT-NMF and (b) BC-SbT-NMF for different numbers of spatial and temporal modules for the natural movie stimulus. Optimal numbers are marked by red squares.

