# OpenReview forum: "Baseline-corrected space-by-time non-negative matrix factorization for decoding single trial population spike trains"
_ICLR.cc/2018/Conference — Reject_

### Official Review · AnonReviewer2 · 2017-11-27
**A baseline subtracted Tucker2 modeling approach with non-negativity to model time binned spike trains of recordings of neural activity. The paper is technically sound but the work appears somewhat incremental.**

**Rating:** 4
**Confidence:** 4

**Review:**

This study proposes the use of non-negative matrix factorization accounting for baseline by subtracting the pre-stimulus baseline from each trial and subsequently decompose the data using a 3-way factorization thereby identifying spatial and temporal modules as well as their signed activation. The method is used on data recorded from mouse and pig retinal ganglion cells of time binned spike trains providing improved performance over non-baseline corrected data.

Pros:
The paper is well written, the analysis interesting and the application of the Tucker2 framework sound. Removing baseline is a reasonable step and the paper includes analysis of several spike-train datasets. The analysis of the approaches in terms of their ability to decode is also sound and interesting.

Cons:
I find the novelty of the paper limited:
The authors extend the work by (Onken et al. 2016) to subtract baseline (a rather marginal innovation) of this approach. To use a semi-NMF type of update rule (as proposed by Ding et al .2010) and apply the approach to new spike-train datasets evaluating performance by their decoding ability (decoding also considered in Onken et al. 2016).

Multiplicative update-rules are known to suffer from slow-convergence and I would suspect this also to be an issue for the semi-NMF update rules. It would therefore be relevant and quite easy to consider other approaches such as active set or column wise updating also denoted HALS which admit negative values in the optimization, see also the review by N. Giles
https://arxiv.org/abs/1401.5226
as well as for instance:
Nielsen, Søren Føns Vind, and Morten Mørup. "Non-negative tensor factorization with missing data for the modeling of gene expressions in the human brain." Machine Learning for Signal Processing (MLSP), 2014 IEEE International Workshop on. IEEE, 2014.

It would improve the paper to also discuss that the non-negativity constrained Tucker2 model may be subject to local minima solutions and have issues of non-uniqueness (i.e. rotational ambiguity). At least local minima issues could be assessed using multiple random initializations.

The results are in general only marginally improved by the baseline corrected non-negativity constrained approach. For comparison the existing methods ICA, Tucker2 should also be evaluated for the baseline corrected data, to see if it is the constrained representation or the preprocessing influencing the performance. Finally, how performance is influenced by dimensionality P and L should also be clarified.

It seems that it would be naturally to model the baseline by including mean values in the model rather than treating the baseline as a preprocessing step. This would bridge the entire framework as one model and make it potentially possible to avoid structure well represented by the Tucker2 representation to be removed by the preprocessing.



Minor:
The approach corresponds to a Tucker2 decomposition with non-negativity constrained factor matrices and unconstrained core - please clarify this as you also compare to Tucker2 in the paper with orthogonal factor matrices.

Ding et al. in their semi-NMF work provide elaborate derivation with convergence guarantees.  In the present paper these details are omitted and it is unclear how the update rules are derived from the KKT conditions and the Lagrange multiplier and how they differ from standard semi-NMF, this should be better clarified.

---

> ### Author Response · Authors · 2018-01-02
> **Rebuttal**
>
> Thank you very much for this detailed review. Below, we reply to your comments and questions:
> 1) To unravel the spatiotemporal activity patterns of ganglion cells during visual stimulation, we need to investigate their spatial and temporal structure. Onken et al. 2016 applied a three factor non-negative matrix factorization with full non-negativity constraints to decompose neural activity recorded from isolated salamander retinas into their non-negative spatial, temporal and activation coefficients. Salamander retinal ganglion cells have very low spontaneous activity. For this reason, the authors could ignore the baseline in that study. In order to decompose neural activity recorded from mouse and pig retina which have non-negligible spontaneous activity, we need a factorization method to deal with negative elements of baseline corrected input signals to obtain biologically interpretable spatial and temporal modules. For this purpose, we extended two factor semi-NMF presented in Ding et al. (2010) to a trial-based tri-factorization where two of the factors are non-negative and trial independent and refer to combinations of neurons firing together and temporal activation of these groups of neurons and one factor is signed and trial dependent and refers to strengths of recruitment of such neural patterns on each trial. We showed that in situations where the stimulus elicits a strong change in ﬁring rate, our extensions yield a better decoding performance compared to SbT-NMF.
> 2) According to N. Gillis. 2014 and Nielsen et al. 2014, multiplicative update rules suffer from slow convergence for image processing, text mining and hyperspectral images, while scaling well for sparse matrices. Matrices of discretized spiking activity of populations of neurons are definitely sparse and consequently multiplicative update rules do seem appropriate for these data. In our study, single trial spike trains of mouse and pig ganglion cells were binned into 10 ms time intervals and decomposed by multiplicative update rules over 1000 iterations. We investigated convergence speed and found that the update rules converged quickly in the first iterations (new Suppl. Fig. S4, page 13). Considering the fast convergence of our multiplicative update rules and their simple implementation, we did not use any alternative technique.
> 3) We addressed your concern regarding local minima by considering 50 different BC-SbT-NMF decompositions with random initializations of spatial and temporal modules and activation coefficients. We selected the decomposition with the lowest reconstruction error and compared decoding performance when using this decomposition with the one that we initially obtained from a single decomposition (Fig. S5, page 14, shown only for the natural movie stimuli). We found that the decoding performance of the multiple initialization decomposition is slightly higher than SbT-NMF. Nevertheless, the very small improvement did not justify the 50-fold increase in computational cost.
> 4) We now considered additional methods to compare against. We modified Suppl. Fig. S2 (page 11) to compare to the decoding performance of Orthogonal Tucker-2, spatiotemporal PCA and spatiotemporal ICA on baseline corrected data and non-baseline corrected data. It is evident that baseline correction improves decoding performance of BC-SbT-NMF, BC-Tucker-2 and BC spatiotemporal PCA, especially in Step of light (SoL) and also Low Contrast flicker (LC) stimuli.
> We also evaluated decoding performance of the decomposition methods by considering different combinations of the number of spatial (P) and temporal (L) modules. We considered a range of the number of spatial modules from 1 to the number of neurons (the latter corresponding to no dimensionality reduction for the spatial dimension) and number of temporal modules from 1 to the number of discretized time points per trial (the latter corresponding to no dimensionality reduction for the temporal dimension). We found that the number of spatial and temporal modules did affect decoding performance. For example, Suppl. Fig. S6 (page 14) shows that BC-SbT-NMF achieved higher decoding performance with lower number of spatial and temporal modules compared to SbT-NMF. The red rectangle in Suppl. Fig. S6 (a) indicates that SbT-NMF can achieve similar decoding performance in this case, but only after increasing the number of spatial and temporal modules. In the comparisons in the main text, we used the optimal number of spatial and temporal modules for each method.
> 5) Inclusion of the baseline in a joint factorization-decoding framework would require a major change of the whole method. While promising, we feel that this would be beyond the scope of this paper. We now discuss this direction in Section Discussion (page 9) and in response to reviewer 2.
> 6) We now clarified the relation of SbT-NMF and BC-SbT-NMF to the Tucker-2 decompositions in Section 2.1 (paragraph 3) and In Section 2.2 (paragraph 1 and 3).

---

### Official Review · AnonReviewer1 · 2017-11-27
**Review of "BC SbT NMF"**

**Rating:** 6
**Confidence:** 3

**Review:**

In this paper, the authors present an adaptation of space-by-time non-negative matrix factorization (SbT-NMF) that can rigorously account for the pre-stimulus baseline activity. The authors go on to compare their baseline-corrected (BC) method with several established methods for dimensionality reduction of spike train data.

Overall, the results are a bit mixed. The BC method often performs similarly to or is outperformed by non-BC SbT-NMF. The authors provide a possible mechanism to explain these results, by analyzing classification performance as a function of baseline firing rate. The authors posit that their method can be useful when sensory responses are on the order of magnitude of baseline activity; however, this doesn't fully address why non-BC SbT-NMF can strongly outperform the BC method in certain tasks (e.g. the step of light, Fig. 3b). Finally, while this method introduces a principled way to remove mean baseline activity from the sensory-driven response, this may also discount the effect that baseline firing rate and fast temporal fluctuations can have on the response (Destexhe et al., Nature Reviews Neuroscience 4, 2003; Gutnisky DA et al., Cerebral Cortex 27, 2017).

---

> ### Author Response · Authors · 2018-01-02
> **Rebuttal**
>
> We thank you for the positive assessment of our work.
> Regarding your mixed results concern, unfortunately, we could not identify any data characteristics that would explain why SbT-NMF outperforms BC-SbT-NMF in certain visual tasks with lower baseline activity such as the step of light stimulus protocol. We emphasize, however, that there are not many such cases and that overall, BC-SbT-NMF outperforms SbT-NMF. Indeed, as you point out, in some situations the baseline can have an advantageous effect on the representation of neural responses, and this might be the case in the few visual tasks were SbT-NMF outperforms BC-SbT-NMF.
> We now discuss this possibility in the Discussion Section (page 9):
> “While BC-SbT-NMF outperformed SbT-NMF overall on tasks with strong baseline activity, we also found that in a few cases, SbT-NMF performed better than BC-SbT-NMF. Previous studies showed that there is an effect of the baseline firing rate on the response (Destexhe et al., Nature Reviews Neuroscience 4, 2003; Gutnisky DA et al., Cerebral Cortex 27, 2017). In these situations, the baseline might have an advantageous effect on the representation of neural responses and could lead to better decoding performance of SbT-NMF that we observed in some cases. One possibility to take this effect into account would be to devise a joint factorization-decoding framework that explicitly introduces the baseline into the optimization framework. While this is beyond the scope of the current work, we believe that development of such a framework is a promising direction for future research.“

---

### Official Review · AnonReviewer3 · 2017-11-28
**Review of baseline-corrected space-by-time non-negative matrix factorization for decoding single trial population spike trains**

**Rating:** 6
**Confidence:** 3

**Review:**

In this contribution, the authors propose an improvement of a tensor decomposition method for decoding spike train. Relying on a non-negative matrix factorization, the authors tackle the influence of the baseline activity on the decomposition. The main consequence is that the retrieved components are not necessarily non-negative and the proposed decomposition rely on signed activation coefficients. An experimental validation shows that for high frequency baseline (> 0.7 Hz), the baseline corrected algorithm yields better classification results than non-corrected version (and other common factorization techniques).

The objective function is defined with a Frobenius norm, which has an important influence on the obtained solutions, as it could be seen on Figure 2. The proposed method seems to provide a more discriminant factorization than the NMF one, at the expense of the sparsity of spatial and temporal components, impeding the biological interpretability.  A possible solution is to add a regularization term to the objective function to ensure the sparsity of the factorization.

---

> ### Author Response · Authors · 2018-01-02
> **Rebuttal**
>
> We thank you for the positive assessment of our work and for the regularization suggestion.
>
> We addressed this suggestion in the revised manuscript in Section 2.2 (pages 3-4) by introducing L1-regularization terms for the spatial and temporal modules in the objective function of the BC-SbT-NMF derivation following the method outlined in Hoyer, JMLR 2004. For simplicity, we considered just one regularization parameter instead of two separate regularization parameters for the spatial and temporal modules. In total, we considered a range of seven values for the regularization parameter: 0, 1, 10, 100, 1000, 10000, 100000. We now included a new Suppl. Fig. S3 (page 12), showing much sparser spatial and temporal modules that we obtained from our L1-regularization in conjunction with BC-SbT-NMF. However, we found that decoding performance decreased for all non-zero L1-regularizations that we applied (Supplementary Figure S3 panel a). Therefore, in the main paper, we report results for the regularization parameter set to 0 (corresponding to the original algorithm) which achieved highest decoding performance, and now mention in Section 2.3 (page 5) that L1 sparsity constraints for BC-SbT-NMF spatial and temporal modules decrease decoding performance.
>
> We hope that Figure S3 as well as the modified derivation address your concern regarding sparsity of spatial and temporal modules for BC-SbT-NMF. Please let us know if you have further comments or questions.

---

### Decision · Program_Chairs · 2018-01-29
**ICLR 2018 Conference Acceptance Decision**

**Decision:**

Reject

**Comment:**

This work is incremental compared to previous work, solving very specific challenges, and would probably appeal to only a very limited fraction of ICLR's audience.